# Environmental Correlates of Motor Competence in Children—The Skilled Kids Study

**DOI:** 10.3390/ijerph16111989

**Published:** 2019-06-04

**Authors:** Donna Niemistö, Taija Finni, Eero A. Haapala, Marja Cantell, Elisa Korhonen, Arja Sääkslahti

**Affiliations:** 1Faculty of Sport and Health Sciences, University of Jyväskylä, 40014 Jyväskylä, Finland; taija.finni@jyu.fi (T.F.); eero.a.haapala@jyu.fi (E.A.H.); leaelisa.korhonen@gmail.com (E.K.); arja.saakslahti@jyu.fi (A.S.); 2Physiology, Institute of Biomedicine, School of Medicine, University of Eastern Finland, 70211 Kuopio, Finland; 3Department of Special Educational Needs and Child Care, University of Groningen, 9712 Groningen, The Netherlands; m.h.cantell@rug.nl

**Keywords:** children, motor competence, environment, geographical location, residential density, outdoor time, participation in sports

## Abstract

Environment, physical activity (PA) and motor development are tightly interwoven during childhood. We examined the associations of environmental factors with motor competence (MC) in children. Children (*N* = 945, 50.1% boys, age = 3–7 years, mean = 5.4 years) from 37 childcare centres in the Southern (*n* = 17), Central (*n* = 13) and Northern Finland (*n* = 7) participated. The environmental factors comprised the geographical location (Southern, Central and Northern Finland) and residential density (metropolitan area, city, rural area and countryside) of the childcare centres’ based on postal codes and the national population density registry. MC was measured using the Test of Gross Motor Development (TGMD)-3, as well as by quantifying time spent outdoors and participation in organised sports via parental questionnaire. It was found that children from the countryside had better MC and spent most time outdoors, while children from the metropolitan area most frequently engaged in organised sports. Gender comparisons revealed that girls outperformed boys in locomotor skills, while boys were better in object control skills, had higher TGMD-3 score and spent more time outdoors. Time spent outdoors and participation in organised sports were associated positively with MC, but not in children from the countryside. In conclusion, higher population density was associated with lower MC and less time spent outdoors. The findings suggest that versatile outdoor environments may support motor development through PA.

## 1. Introduction

Motor competence (MC) enables children to participate in various physical activities (PA) and physically active play [1,2]. Better MC has also been found to predict children’s subsequent PA levels [3]. However, the evidence from previous studies suggests negative secular changes in MC over the past decades [4,5], and children struggle to achieve the recommended levels of daily PA [6]. Stodden et al. [1] proposed a bidirectional connection between MC and PA. Furthermore, MC influence children’s motivation and engagement in PA [1,2]. Therefore, the declining MC in children may be due to their decreasing PA [5] or changes in their living environment [7,8].

According to Gibson [9], the theory of affordances refers to the functionally significant properties of the environment; he describes the variety of affordances an environment offers that animals (i.e., terrain, shelters and surfaces) can use for multiple purposes. Gibson [9] defines affordance as a combination of physical properties of the environment that fit one’s actions and locomotor systems. The theory of affordance has also been applied to young children’s motor learning contexts [10] as well as explaining PA in children, in terms of outdoor play [11], independent mobility [11,12] and the amount of affordances in the environment [13]. Previous research has shown that children find outdoor environments stimulating and motivating [12,14,15], for example, large yards that provide affordances to play and run [16]. Indeed, free running and playing are important for the development of locomotor (LM) skills, such as walking, running, climbing, galloping and jumping [17]. Furthermore, large spaces and areas of play are also crucial to practise object control (OC) skills [18].

Overall, MC development is stimulated through appropriate challenges to maintain balance, achieve objectives and move from the current place to another location. Campos et al. [19] suggested that improved LM skills provide more numerous and variable social and cognitive experiences and support the development of such experiences during infancy. After infancy, LM skills may be responsible for an enduring role in development by maintaining and updating existing skills. Such development is possible when children have social support and psychological freedom to move in an environment with interesting affordances [16]. During early childhood, handling different objects opens up new opportunities for visual, manual, and oral exploration [20]. This is significant because good hand/foot-eye coordination is essential to the development of OC skills [21,22]. Besides LM and OC skills, also climbing and balancing are essential for children’s MC development [23,24]. All the aforementioned skills serve to develop children’s MC, at both low and high PA intensities. However, as discussed above, to practise for example walking or running there is a need for divergent affordances in addition to climbing, throwing or balancing. This demonstrates that environment and its affordances can contribute to motor development and PA.

Motor development not only influences the PA level, but is also related to cognitive functions. Recently, MC has been associated with cognitive functions, such as executive functions [21], attention and working memory, and information processing speed [25,26,27,28] as well as language development [29] and reading [30]. These associations have been explained through the adaptations of brain structures and functions [31]. Therefore, to support cognitive functions, the environment has an important role providing affordances to practise motor skills. On the other hand, for motor learning the perfect timing for the child to learn skills is essential and thus recognised as a zone of proximal learning [32]. When a child has the prerequisites to learn a new skill but is not yet able to do it alone, the environment or other people (peers or adults) can support and scaffold this appropriate level. 

As previously stated, a versatile outdoor environment can offer a proximal zone of development. A physically activating environment is demonstrated to be safe and to offer the possibilities for free play [33]. Secondly, adults can create a proximal zone to support MC learning through organised physical activities for children. Organised sports have been shown to increase children’s PA [8], and frequent participation in organised sports has been found to improve MC [34]. Furthermore, early educators in the childcare centres can influence the role of curriculum, which is crucial to defining age appropriate contents, amount, quality and environments to support motor skill learning [34]. Children are motivated to be active and practise new MC when they feel themselves to be competent, have the autonomy to choose appropriate activities and are able to be with other children [35]. Therefore, from a proximal learning perspective, the timing of different motor challenges during children’s development becomes meaningful. It follows that child falls out of zone if he already is competent or he does not yet have abilities to perform the task [32]. Thus, the more variation an environment provides, the more there is the potential to offer appropriate and timely challenges. For this reason, it is assumed that the physical environment plays an important role in children’s MC development, which by extension supports their cognitive functions. 

From a global perspective, countries have widely different living environments, which is likely to cause differences in MC via diversity in socio-cultural and geographical aspects [36]. While some cross-country comparisons of children’s MC are available [4,5,37,38], the effects of environmental differences within a single country are less studied. Therefore, information about how environmental factors within one country are associated with MC in children is scarce, particularly with regard to time spent outdoors and participation in organised sports. The environmental factors referred to in this study are geographical location of the living place (Southern, Central and Northern Finland) and residential density (metropolitan area, cities, rural areas and countryside). We assume that the amount of daylight and the mean temperature present in each geographical location may have a mediating role to time spent outdoors, because previous studies suggested that children tend to be less physically active in cold seasons [39,40,41]. We were interested in learning if this phenomenon can be observed in relation to MC. It is also important to study residential density because the MC of young children seems to be declining [4] simultaneously with a global population trend towards living in bigger cities [42]. We hypothesised that less inhabited areas, contain more natural, unbuilt parks that include several different landforms provide venues for children to practise balance and coordination [16], and therefore children in rural areas and countryside may have better MC. By contrast, cities and denser areas contain parks and playing areas that include fixed equipment, such as slides, climbing bars, jungle gyms and tunnels that allow children to practise mainly balancing and strength-demanding skills [17,23]. However, we question if that is enough to develop MC. Finally, we hypothesised that children living in the rural areas and countryside spend more time outdoors and have fewer opportunities to participate in organised sports than children living in cities or metropolitan areas. In one Nordic country, we examined whether children’s physical living environment i.e., geographical location and residential density, are associated with their MC, the amount of time spent outdoors and participation in organised sports. 

## 2. Methods

### 2.1. Study Protocol and Study Participants

The Skilled Kids study protocol and cluster-random sampling have been previously described in detail [43,44]. Briefly, we aimed to recruit a geographically representative sample of 1000 children, aged 3–7 years, from Finnish childcare centres. The Finnish national registry of early educators includes 2600 childcare centres. Based on this register, childcare centres were chosen using cluster-random sampling from the Southern, Central and Northern Finland based on postal codes. Of the 47 childcare centres that we invited to participate in the study, 37 centres agreed to participate, with a total of 945 children, aged over 36 months (mean age = 5.4 years, 473 boys or 50.1%) and with complete data on the Test of Gross Motor Development—third edition (TGMD-3). Of the childcare centres, 17 were located in Southern, 13 in Central and 7 in Northern Finland (Table 1).

The Ethics Committee of the University of Jyväskylä, Finland, granted approval for the study on 31 October 2015 (Skilled Kids, 31.10.2015). The parents of the participating children provided their written consent. The children were informed about all study procedures and their right to opt out of participation at any time, without consequences.

### 2.2. Physical Environment: Geographical Location and Residential Density

Both geographical location and residential density were evaluated indirectly by using the set of postal codes of the childcare centres that the children were attending as the reference and the national population density registry for the categorisation. Finland was divided into three geographical locations: Northern, Central and Southern Finland. Additionally, as the residential density might affect the possibilities for the children’s time spent outdoors and for organised sports, the rest of the country was classified according to residential density, comprising four categories: the metropolitan area, cities, rural areas and the countryside (Table 1).

### 2.3. Motor Competence

MC was measured with the TGMD-3 [45], which has two skill categories: LM skills and OC skills. LM skills constitute a summary of six skills evaluated by points, as follows: run (0–8 points), gallop (0–8 points), hop (0–8 points), skip (0–6 points), horizontal jump (0–8 points) and slide (0–8 points), for a maximum total of 46 points. OC skills include a summary of seven skills, as follows: two-hand strike of a stationary ball (0–10 points), one-hand forehand strike (0–8 points), one-hand stationary dribble (0–6 points), two-hand catch (0–6 points), kicking a stationary ball (0–8 points), overhand throw (0–8 points) and underhand throw (0–8 points), for a maximum total of 54 points. An educated observer, who analysed the skills according to the fulfilment of the given criteria (3 to 5 criteria for one skill), evaluated each skill (0 point if the given criteria were not fulfilled, 1 point if they were met). Each child performed each skill twice, and his/her evaluation score was the sum of the received points during these two performances. The TGMD-3 total score was the sum of LM and OC skills, with a theoretical maximum of 100 points.

The TGMD-3 protocol was carefully followed according to the manual and described previously [43,44]. The TGMD-3 has been demonstrated to have good to excellent intrarater and interrater reliability [45], and it has been found valid and reliable both internationally [46,47] and in this study’s context [43,44]. With this sample, the interrater reliability for the TGMD-3 total skills was 0.88 (95% confidence interval [CI] = 0.85–0.92), tested among 167 children [44].

### 2.4. Time Spent Outdoors and Participation in Organised Sports

The time spent outdoors and participation in organised sports were assessed through a questionnaire administered to the parents. The data on the time spent outdoors were obtained by asking this question: “How much time, on average, does your child spend outdoors after a preschool (or childcare) day/on the weekends?” The scale for the weekdays ranged from 0 to 3 (0 = not at all, 1 = under 30 min/d, 2 = approximately 30–60 min/d and 3 = over 60 min/d), and the scale for the weekends ranged from 0 to 4 (0 = not at all, 1 = under 30 min/d, 2 = approximately 30–60 min/d, 3 = 1–2 h/d and 4 = over 2 h/d). The total score from both scales was used to represent the time spent outdoors. The question’s test-retest reliability was investigated with 30 responses and found to be appropriate (ICC = 0.62; 95% CI = −0.12–1.0). The data on participation in organised sports were obtained by asking this question: “Does your child participate in organised PA or sports in a group or a sports club?” If the answer was “yes”, further questions regarding such activities were asked, as follows: “How many times a week?” “For how many minutes at a time?” The total time (in minutes) spent on organised sports per week was calculated and used in the analyses. The test-retest reliability of the questions was analysed and found to be good (ICC = 0.81; 95% CI = 0.60–0.91).

### 2.5. Other Assessments

The children’s height, weight, body mass index standard deviation scores (BMI-SDS) [48] and their parents’ educational and income levels were assessed, as described in detail previously [43,44] (Table 2).

### 2.6. Statistical Analysis

IBM SPSS version 24.0 (IBM Corp., Armonk, NY, USA) was used for the analyses, and the level of significance was set at *p* < 0.05. Descriptive statistics (mean and standard deviation [SD]) were calculated for MC (LM, OC and TGMD-3 skills) and for time spent outdoors and participation in organised sports. Furthermore, because previous studies [49,50] and the present study (Table 2) showed differences in MC between boys and girls, further analyses were performed separately for girls and boys. 

The correlations between MC (LM, OC skills and TGMD-3), time spent outdoors and participation in organised sports was analysed with partial correlation adjusting for age in months. To analyse the differences between gender, geographical locations (Southern, Central and Northern Finland) and residential density (metropolitan area, cities, rural areas and countryside) in MC (LM, OC skills and TGMD-3), the time spent outdoors and participation in organised sports, the linear mixed-effects model was used (Table 3, Table 4, Table 5 and Table 6). The age and the random effect for childcare centres were adjusted for the model. LM, OC skills, TGMD-3 total score, time spent outdoors and participation in organised sports were used as separate outcome variables, using geographical location and residential density as categorical explanatory variables one at a time. The effect of childcare centre was decided to adjust for random effect as the previous study [51] with this dataset showed that childcare centres’ were associated with MC in children, which corresponds with the findings of several other studies regarded to MC or PA [10,52].

## 3. Results

### 3.1. Descriptive Results

All 945 children’s ages ranged from three to seven years (mean = 5.4 years, SD = 1.1). About half of the children were boys (*n* = 473 or 50.1%). The parents who answered to the questionnaire (*n* = 936; mean age = 35.8 years, SD = 5.4) were most likely mothers (*n* = 816 or 87.2%), and more than half of the parents had polytechnic- or university-level education (*n* = 569 or 60.7%). Nearly every child spoke Finnish as his/her mother tongue (*n* = 886 or 94.5%). The majority of the children lived in cities, the metropolitan area, or Central or Southern Finland (Table 1). The parents living in the metropolitan area had the highest annual income (*n* = 172) and the highest educational level (*n* = 187), while the parents living in cities (*n* = 386) and the countryside (*n* = 134) had the lowest annual income. The parents living in the countryside had the lowest educational level (*n* = 150) (Table 1). Using the total sample, we found that the TGMD-3 total score correlated with LM skills (r = 0.83; *p* < 0.001), OC skills (r = 0.0.84; *p* < 0.001), time spent outdoors (r = 0.12; *p* < 0.001) and participation in organised sports (r = 0.23; *p* < 0.001). LM skills correlated with OC skills (r = 0.38; *p* < 0.001), time spent outdoors (r = 0.07; *p* = 0.029) and with participation in organised sports (r = 0.14; *p* < 0.001). OC skills correlated with time spent outdoors (r = 0.13; *p* < 0.001) and finally, participation in organised sports (r = 0.23; *p* < 0.001). 

### 3.2. Gender Differences

The descriptive data of the study sample are reported in Table 2. Boys had lower LM skills, better OC skills and a better TGMD-3 total score than girls (Table 2). Boys also spent more time outdoors.

### 3.3. Geographic Location

The girls from Central Finland spent more time outdoors than girls from the Southern part of the country, while girls from the Southern area participated more in organised sports than girls from Central Finland (Table 3). Additionally, time spent outdoors correlated positively with LM skills in Central Finland (r = 0.21; *p* = 0.007). Participation in organised sports correlated positively with LM skills in Central (r = 0.20; *p* = 0.01) and Northern Finland (r = 0.23; *p* = 0.05) and with better OC skills in Southern (r = 0.15; *p* = 0.04) and Northern Finland (r = 0.39; *p* < 0.001).

The boys from Central and Northern Finland spent more time outdoors than boys living in the Southern part of the country (Table 4). Moreover, time spent outdoors correlated positively with OC skills in Southern Finland (r = 0.14; *p* = 0.04). Participation in organised sports correlated positively with LM skills in Southern (r = 0.16; *p* = 0.02) and in Central Finland (r = 0.20; *p* = 0.01) while in Northern Finland the correlation with LM skills was negative (r = −0.33; *p* = 0.005). Participation in organised sports correlated positively with OC skills in Southern (r = 0.41; *p* < 0.001) and Central Finland (r = 0.20; *p* = 0.01).

### 3.4. Residential Density

Girls living in the countryside outperformed other girls in LM, OC skills and in TGMD-3 total scores. Additionally, girls from metropolitan area had better LM skills than girls from rural areas, while girls from rural areas had better OC skills and TGMD-3 total scores than girls from metropolitan area. Girls from the countryside spent more time outdoors than girls from metropolitan or rural areas. Girls from metropolitan area participated more in organised sports than girls from cities or rural areas. Finally, girls from the countryside participated more in organised sports than girls from rural areas (Table 5). Among girls, participation in organised sports correlated positively with LM skills (r = 0.25; *p* = 0.02) in metropolitan area and with LM (r = 0.15; *p* = 0.03) and OC (r = 0.21; *p* = 0.03) skills in cities.

Boys from the countryside outperformed the boys from the rural areas in LM skills. Boys from the countryside scored higher in OC skills than boys from the metropolitan area. The boys from the countryside had a higher TGMD-3 total score than the boys from the rural areas. The boys from metropolitan area spent less time outdoors compared with boys from the cities and the countryside. Additionally, boys living in rural areas spent less time outdoors than boys living in the countryside. (Table 6). Time spent outdoors correlated positively with LM skills in cities (r = 0.17; *p* = 0.02) and with OC skills in metropolitan area (r = 0.23; *p* = 0.03). Participation in organised sports correlated positively with OC skills in metropolitan area (r = 0.30; *p* = 0.004), in cities (r = 0.38; *p* < 0.001) and in rural areas (r = 0.26; *p* = 0.02).

## 4. Discussion

In this study, we investigated the associations between geographical location and the residential density of the living environment with MC, time spent outdoors and participation in organised sports in 3–7-year-old Finnish children. The main finding was that residential density was more strongly associated with MC, time spent outdoors and participation in organised sports than geographical location was. Specifically, girls from the countryside had better MC than their peers from metropolitan area, cities or rural areas. Girls living in metropolitan area engaged more in organised sports than the other girls. Boys from the countryside outperformed boys from rural areas in LM skills and TGMD-3 total scores and boys from metropolitan area in OC skills. Furthermore, in the whole sample of children, some gendered differences emerged. Finally, time spent outdoors and especially participation in organised sports, was associated with better MC. However, similar associations between better MC and time spent outdoors and participation in organised sports were not observed among children from the countryside, despite the fact that they had the highest MC. Thus, we suggest that children with lower MC skills tend to benefit more from time spent outdoors and participation in organised sports. As cognitive functions and MC are positively associated [25,26,29,30], this research is a valuable contribution to the literature regarding the associations among MC, PA and environmental factors.

### 4.1. Geographical Location

We found that children from Central Finland spent the most time outdoors, while children from Southern Finland spent the least time outdoors. Therefore, the mean temperature and the amount of daylight were not significant determinants of outdoor time, which was counter to our hypothesis. Furthermore, we observed that girls who spent more time outdoors had better LM skills in Central Finland, while boys who spent more time outdoors had better OC skills in Southern Finland. In addition, we found no differences in LM, OC skills or MC among children from different geographical locations. This result may reflect the national curriculum of early education [53], which covers the whole nation and supports equal educational actions and recommendations for PA [54] for all children among early education. Furthermore, Finnish children can move around quite freely and independently [11] due to the right of common access to the environment and its affordances. Therefore, it may be that Finnish children have equal opportunities to develop MC, and to participate in organised sports, regardless of geographical location. These findings together suggest that the equal and free access to multiple affordances and participation in organised sports in Finland may greatly benefit motor learning in children. This is in line with the theory of affordances, which holds that playing outdoors motivates and supports motor development [14]. Coté and his research group [55,56] have shown that the majority of professional athletes grew up in small cities, which offer more equal possibilities for free play and access to organised sports [57]. As a whole, our data suggests that early childhood years are important to development of MC; free play provides children possibilities to feel autonomy and freedom to play [35], while access to organised sports also contributes to the development of motor learning in children. Those factors, together with multiple environmental affordances [14], are strong motivators for PA and motor development. 

Interestingly, the geographic characteristics of Northern Finland (a long dark period with low temperature) are not disadvantages in terms of the time children spent outdoors or their motor development. Some previous studies have shown an inverse association between the temperature and PA levels [39,40,41] but the findings with regard to younger children (less than 8-years) are inconsistent [40]. In Finnish context, Soini et al. [58] found that season only minimally influence children’s PA levels, and that other factors (e.g., gender, educational support by parents and teachers) are more significant correlates of PA and motor development in children who are less than 4-years old.

Finally, we found that children from Central Finland tend to spend the most time outdoors. Central Finland has many small cities, lots of unbuilt spaces and nature elements (e.g., hills, lakes and forests) around people’s everyday living environment. According to previous studies [7,8], varying surfaces and shapes, such as natural and built playground facilities, not only increase children’s PA but also support their motor development. Therefore, it is suggested that versatile environments create proximal zones of development [32] for MC learning. Additionally, to motivate children to move, it is fundamental to hear what they want from their environment [15]. Children typically want their living environment to have active and natural spaces with natural elements, such as water, animals, stones, leaves, sand and sticks [15], all of which are present in Central Finland.

### 4.2. Residential Density

We found that the children from the countryside (with the lowest residential density) had better MC and spent larger amounts of time outdoors than their peers from metropolitan area (with the highest residential density), especially among girls. Girls living in the countryside outperformed other girls in LM, OC skills and in TGMD-3 total scores. Boys from the countryside outperformed boys from rural areas in LM skills and TGMD-3 scores and scored higher in OC skills than boys from the metropolitan area. 

One reason that children from countryside succeed in MC may be access to large spaces and freedom to move, which increases the children’s PA and their development of MC. Kyttä [11] stated that Finnish children have more freedom than their peers from Western Europe do and that less dense areas may provide better possibilities for independent mobility. This suggests that for Finnish children, the freedom of independent mobility increases the pleasure derived from PA. Additionally, the amount of time spent outdoors, which was higher in the countryside than in other regions, has been shown to be positively associated with children’s PA levels [59], which could partly explain the better MC demonstrated by children from the countryside. 

With regard to MC, Campos et al. [19] proclaimed that LM skills are fundamental for future MC learning; it may be that if PA levels remain low due to the lack of space or safety, LM skills do not develop as much, and OC skill learning can be delayed. Children from the countryside may have more space and time to repeat those motor skills that are critical for them during that developmental phase. Moreover, playing ball games requires large, empty places, which are usually lacking in a metropolitan area [18] resulting lower level of OC skills in children from metropolitan area. 

As PA and motor development are associated with each other [1,2], the possibility to move freely in less densely populated areas in everyday life may be associated with better MC skills or more time spent outdoors, as demonstrated by our sample of children from the countryside. In fact, a previous study [60] showed that children’s PA participation in everyday life was positively associated with publicly provided recreational infrastructure, such as access to recreational facilities and schools, and transport infrastructure, such as the presence of sidewalks and controlled intersections, access to destinations and public transportation. On the other hand, in densely populated areas, parents may exert more control or restriction of their children’s time spent outdoors due to the lack of safety [60,61] although in general Finland is perceived as a relatively safe environment [61]. 

Children seem to prefer versatile environments near home [61] that provide large, safe spaces with natural elements that encourage the development of LM, OC skills and balance skills. In line with the theory of affordances [9], it seems that the more variation the environment and affordances provides, the more possibilities the child may have for divergent motor learning. Thus, the result is two-fold: the variety of living environments may be greater in less dense areas, which explains why children from the countryside display more advanced motor skills, and secondly, tend to spent more time outdoors.

*Childcare centre.* In the present study, even though Finnish children’s equality in terms of geographical location was evident, there were differences between childcare centres in MC. A previous study [51] with this study sample showed that childcare centres that had large yards, with variation in shapes and the amount of surfaces, were positively associated with children’s MC [51]. Thus, it is significant that childcare centres with large yards are more common in less densely populated areas, like in the countryside. Similarly, a study by Kyttä [13] showed that in Finland, areas with lower population density provided the largest number of actively available affordances, while areas with high population density had the lowest number of affordances, although her focus was not specifically on childcare centres. These findings are in accord with the theory of affordances, which proclaims that motor development benefits from environmental affordances [14] located in children’s everyday living environment [61]. Because children spend multiple hours in childcare centres, we believe that the environment near these centres plays a notable role in motor development. Children from the countryside spent more time outdoors at home than children from metropolitan (girls and boys) or rural areas (girls) and they benefit from childcare centres larger yards, so they may have more opportunities to develop their motor skills both at the childcare centre and at home, which would explain their better MC.

*Participation in organised sports.* In the present study, the metropolitan children, especially the girls, participated the most in organised sports, as expected. Contrary to our hypotheses, children from the countryside were not disadvantaged by their lack of participation in organised sports. However, children with lower MC, participation in organised sports can be crucial for their MC learning, as MC does not develop optimally with increasing age and maturity but needs to be practised and reinforced continuously during childhood. Previous studies [17,32] suggested that skilled adults’ guidance in organised sports could support children’s MC learning. Additionally, Brian et al. [4] state that the development of OC skills is heightened by participation in specific contexts where children receive accurate instructions. Similarly, the present study revealed that a higher level of participation in organised sports was positively associated with MC in rural areas, cities and metropolitan area. However, no such finding was found with children from the countryside. Thus, we suggest that children, who were not from the countryside, who have lower MC overall, benefitted from participation in organised sports. Since the abovementioned associations were not found in children living in the countryside, it is important to examine further, which other contextual factors could play a mediating or moderating role in children’s participation in organised sports.

### 4.3. Gender Differences

Girls outperformed boys in LM skills, but boys outperformed girls in OC skills and in the TGMD-3 total score. Additionally, boys spent more time outdoors. These findings are in line with those of previous studies [18,62], as boys are typically recognised as having better OC skills than girls. This result may reflect the content of gender differences in play, as girls participate more in organised sports involving LM skills, such as dance [63], while boys engage more in hobbies that include mastery of ball skills [21]. Some researchers suggest that environmental and socio-cultural factors may be the reason for gender differences in children’s OC skills [18,64]. According to some researchers [65], boys are more social and significantly more likely to be involved in ball games, while girls are more likely to play in smaller groups, involving more conversation, sedentary play, jump-skipping and verbal games. These differences may reflect the gender differences in motor development as well.

### 4.4. Suggestions for Future Research from the Cognitive Development Perspective

Our findings indicate that the environment matters for children’s motor development. Earlier research has shown close relationships among brain structures, functions and cognitive functions [31], such as executive functions [21], working memory and information processing [28,66]. Controlling balance and body movements in the natural environment and on different surfaces requires constant brain activity. Appropriate stimuli from parents, peers or the environment are required for normal cognitive development. Nevertheless, there is limited evidence about the effects of different types of PA and the time spent outdoors in different environments on cognitive skills or brain development, and more research on this topic is warranted. However, to optimally support motor development, age appropriate psychological motivation is also necessary. This match meets in the proximal zone of development [32]. In future studies, also from the cognitive development perspective, it would be important to investigate interactions and combined effects of MC, PA as well as time spent outdoors and participation in organised sports on cognitive development.

### 4.5. Strengths and Weaknesses of the Study

This study’s strengths include a relatively large sample of children from varying geographical locations in Finland, and the valid and reproducible methods used to measure MC. We assessed the time spent outdoors and participation in organised sports using a questionnaire administered to parents because a questionnaire is the only feasible method to assess the type and the setting of PA among children. However, it would have been optimal to combine parent-reported measures of PA (time spent outdoors) with device-based measures of PA using accelerometers. Additionally, to gain a deeper understanding of the relationship between cognitive functions and MC, the study would have benefited greatly by measuring cognitive functions. Furthermore, we were unable to assess physical environments other than the childcare centres, and we could not rule out the possibility that local differences in built environments had a small effect on our results. Despite the efforts to include a fully representative sample of Finnish children attending childcare centres, the results revealed a bias towards more highly educated parents. Moreover, it can be assumed that this kind of study interests parents with positive attitudes about a physically active lifestyle. Therefore, our sample may not perfectly represent the Finnish population in different parts of the country.

## 5. Conclusions

The current study provided an example of how children’s daily living environment and MC are closely related in the Finnish context. The main finding revealed that residential density is related to children’s MC, engagement in outdoor play and organised sports. At its best, the daily environment provides children with versatile opportunities to motor learning. Indeed, it was found that Finnish children living in the countryside spent more time outdoors and had higher MC than their age peers in the metropolitan area. The conclusion is that time spent in a physical environment that provides the affordances needed for physical activity is closely related to the development of MC. Furthermore, such an environment enables also learning in a broader perspective; while moving, children perceive and observe their environment. If the environment is safe and engaging enough, children are likely to be fit both physically and cognitively. Therefore, future research can provide more understanding of the multifaceted benefits of physical environments to children’s motor and cognitive learning in variable residential densities and geographical locations.

## Figures and Tables

**Table 1 ijerph-16-01989-t001:** Geographical location, residential density and characteristic of living environment of study sample.

Physical Environment	Localities	Childcare Centres	Educational Level	Income Mean	Children (*n*)			Age in Years (SD)			% of the Study Sample		
**Geographic location (°C ^1^/h/day ^2^)**	***n***	***n***	**Range 1 to 4** **(SD)**	**Range 1 to 8** **(SD)**	**All**	**Girls**	**Boys**	**All**	**Girls**	**Boys**	**All**	**Girls**	**Boys**
Southern (−6.6 to +17.7/6 to 19)	10	17	2.90 (0.83)	4.58 (1.56)	449	224	225	5.30 (1.09)	5.24 (1.07)	5.35 (1.11)	47.5	47.4	47.5
Central (−8.1 to +16.8/5 to 20)	10	13	2.63 (0.74)	4.00 (1.28)	335	163	172	5.59 (1.13)	5.57 (1.18)	5.62 (1.08)	35.4	34.5	36.4
Northern (−11.2 to +15.1/2.5 to 24)	4	7	2.48 (0.70)	4.30 (1.35)	161	85	76	5.40 (1.15)	5.41 (1.15)	5.38 (1.15)	17.1	18.1	16.1
**Residential density** **(n/km^2^)**													
Metropolitan (876.4–2964)	2	6	3.17 (0.79)	4.91 (1.68)	189	94	95	5.17 (1.15)	5.09 (1.08)	5.25 (1.20)	20.0	19.9	20.0
Cities (24.65–762.9)	13	17	2.69 (0.77)	4.15 (1.45)	421	211	210	5.60 (1.11)	5.53 (1.18)	5.66 (1.04)	44.5	44.7	44.4
Rural areas (4.93–64.35)	5	7	2.56 (0.69)	4.31 (1.24)	183	98	85	5.48 (1.15)	5.49 (1.15)	5.47 (1.15)	19.4	20.8	18.0
Countryside (1.49–8.56)	4	7	2.53 (0.69)	4.13 (1.22)	152	69	83	5.15 (0.97)	5.18 (0.91)	5.12 (1.02)	16.1	14.6	17.5
In total sample	24	37	2.73 (0.80)	4.33 (1.46)	945	472	473	5.42 (1.12)	5.38 (1.13)	5.45 (1.11)	100	49.9	50.1

Educational level (1 = comprehensive school; 2 = high school/vocational school; 3 = polytechnic; 4 = university). Income mean (1 = 0 to 13,999 €/year, 2 = 14,000 to 19,999 €; 3 = 20,000 to 39,999 €; 4 = 40,000 to 69,999 €; 5 = 70,000 to 99,999 €; 6 = 100,000 to 119,000 €; 7 = 120,000 to 139,000 €; 8 = over 140,000 €). Values are reported as mean (standard deviation) scores or percentages (%). ^1^ Mean temperature in February (coldest month) and in July (warmest month). ^2^ The amount of daylight in 21st of December (least daylight; winter solstice) and 21st of June (most daylight; summer solstice).

**Table 2 ijerph-16-01989-t002:** The descriptive data of the study sample.

Child Factors	*N*	Mean (SD)	Min	Max	Mean (SD) Girls	Mean (SD) Boys	Gender Differences *p*-Value
Age (years)	945	5.42 (1.12)	3.08	7.75	5.38 (1.13)	5.45 (1.11)	0.36
BMI SDS (%)	943	0.19 (1.05)	−4.55	3.45	0.21 (1.13)	0.17 (0.98)	0.56
Significantly underweight	15	1.6			2.5	0.6	0.66
Underweight	22	2.3			2.2	2.5	0.12
Normal weight	687	72.9			77.1	68.6	<0.001
Overweight	178	18.9			14.4	23.4	<0.001
Obesity	41	4.3			3.8	4.9	0.001
Height (cm)	943	113.52 (9.73)	86.30	137.30	112.64 (10.05)	114.40 (9.33)	0.001
Weight (kg)	943	21.19 (4.47)	11.30	41.60	21.04 (4.73)	21.34 (4.19)	0.62
TGMD-3 locomotor skills (0–46 p.)	945	27.52 (8.07)	0	46	28.89 (7.78)	26.16 (8.13)	<0.001
TGMD-3 object control skills (0–54 p.)	945	24.87 (9.06)	3	50	22.43 (7.91)	27.29 (9.49)	<0.001
TGMD-3 total score (0–100 p.)	945	52.39 (15.16)	4	88	51.32 (14.11)	53.46 (16.08)	0.030
Time spent outdoors (%)	938	100					0.001
Less than 1 h/day	94	10.0			12.8	7.2	
Approximately 1 h/day	493	52.6			52.2	52.9	
1 to 2 h/day	351	37.4			35.0	39.9	
Participation in organised sports (mins/week)	902	49.50 (65.28)	0	421.00	48.34 (59.84)	50.65 (70.27)	0.97

BMI, Body mass index; SDS, standard deviation scores. Motor competence was measured using the Test of Gross Motor Development (TGMD)-3. Time spent outdoors and participation in organised sports were asked via parental questionnaire. Values are reported as mean and standard deviation (SD) scores or percentages (%) and adjusted for age.

**Table 3 ijerph-16-01989-t003:** Geographic location: differences in girls.

Child Factors	Total Sample	Southern			Central			Northern		
	Overall *p*-Value	*n*	Mean (SD)	Adj. Age	*n*	Mean (SD)	Adj. Age	*n*	Mean (SD)	Adj. Age
LM skills (max.46p.)	0.93	224	28.40 (7.81)	28.99	163	29.47 (7.70)	28.71	85	29.05 (7.87)	28.93
OC skills (max. 54p.)	0.27	224	20.83 (7.75)	21.48	163	24.23 (7.78)	23.41	85	23.21 (7.80)	23.08
TGMD-3 total (max. 100p.)	0.58	224	49.23 (13.96)	50.48	163	53.71 (13.90)	52.12	85	52.26 (14.25)	52.00
Time spent outdoors (scale 1–7)	**0.06**	222	4.82 (1.15)	**4.83 ^1^**	162	5.14 (1.24)	**5.12 ^1^**	85	5.08 (1.15)	5.08
Participation in organised sports (mins/week)	**0.03**	208	54.68 (63.05)	**56.98 ^1^**	160	38.23 (52.63)	**35.28 ^1^**	80	52.10 (62.85)	51.82

Motor competence was measured using the Test of Gross Motor Development (TGMD)-3. Time spent outdoors and participation in organised sports were asked via parental questionnaire. The time spent outdoors was a sum from the time spent outdoors on weekdays and weekends. The scale for the weekdays ranged from 0 to 3 (0 = not at all, 1 = under 30 min/d, 2 = approximately 30–60 min/d and 3 = over 60 min/d), and the scale for the weekends ranged from 0 to 4 (0 = not at all, 1 = under 30 min/d, 2 = approximately 30–60 min/d, 3 = 1–2 h/d and 4 = over 2 h/d). Values are reported as mean and standard deviation (SD) scores. LM skills= locomotor skills, OC skills= object control skills. * Statistically significant difference (adjusted for age, random effect of childcare centre) between geographical locations groups at the level of *p* < 0.05. Statistically significant differences in bold. In time spent outdoors difference between Central and Southern: 1 *p* = 0.03 *. Participation in organised sports, difference between Southern and Central: 1 *p* = 0.012 *.

**Table 4 ijerph-16-01989-t004:** Geographic location: differences in boys.

Child Factors	Total Sample	Southern			Central			Northern		
	Overall *p*-Value	*n*	Mean (SD)	Adj. Age	*n*	Mean (SD)	Adj. Age	*n*	Mean (SD)	Adj. Age
LM skills (max.46p.)	0.61	225	26.12 (7.96)	26.58	172	26.33 (8.17)	25.57	76	25.93 (8.64)	26.28
OC skills (max. 54p.)	0.46	225	26.27 (9.33)	26.83	172	29.02 (9.31)	28.10	76	26.42 (9.90)	26.84
TGMD-3 total (max. 100p.)	0.94	225	52.39 (15.61)	53.42	172	55.35 (16.10)	53.67	76	52.36 (17.18)	53.13
Time spent outdoors (scale 1–7)	**0.001**	223	4.82 (1.15)	**4.94 ^1,2^**	170	5.55 (1.10)	**5.53 ^1^**	76	5.36 (1.10)	**5.37 ^2^**
Participation in organised sports (mins/week)	0.26	219	55.87 (76.38)	58.11	163	46.43 (62.79)	42.32	72	44.34 (66.58)	46.85

Motor competence was measured using the Test of Gross Motor Development (TGMD)-3. Time spent outdoors and participation in organised sports were asked via parental questionnaire. The time spent outdoors was a sum from the time spent outdoors on weekdays and weekends. The scale for the weekdays ranged from 0 to 3 (0 = not at all, 1 = under 30 min/d, 2 = approximately 30–60 min/d and 3 = over 60 min/d), and the scale for the weekends ranged from 0 to 4 (0 = not at all, 1 = under 30 min/d, 2 = approximately 30–60 min/d, 3 = 1–2 h/d and 4 = over 2 h/d). Values are reported as mean and standard deviation (SD) scores. LM skills= locomotor skills, OC skills= object control skills. * Statistically significant difference (adjusted for age, random effect of childcare centre) between geographical locations groups at the level of *p* < 0.05. Statistically significant differences in bold. Time spent outdoors, difference between Central and Southern: **1**
*p* = 0.028 * and Northern and Southern: 2 *p* = 0.031 *.

**Table 5 ijerph-16-01989-t005:** Residential density: differences in girls.

Child Factors	Total Sample	Metropolitan			Cities			Rural Areas			Countryside		
	Overall *p*-Value	*n*	Mean (SD)	Adj. Age	*n*	Mean (SD)	Adj. Age	*n*	Mean (SD)	Adj. Age	*n*	Mean (SD)	Adj. Age
LM skills (max.46p.)	0.076	94	27.46 (8.35)	**28.67 ^1,4^**	211	29.24 (7.39)	**28.62 ^2^**	98	28.74 (7.75)	**28.30 ^3,4^**	69	29.94 (8.07)	**30.82 ^1,2,3^**
OC skills (max. 54p.)	**0.054**	94	19.70 (7.84)	**21.05 ^1,4^**	211	22.73 (7.71)	**22.05 ^2^**	98	22.86 (7.25)	**22.36 ^3,4^**	69	24.64 (8.65)	**25.60 ^1,2,3^**
TGMD-3 total (max. 100p.)	0.036	94	47.16 (14.63)	**49.72 ^1,4^**	211	51.98 (13.65)	**50.67 ^2^**	98	51.60 (13.36)	**50.66 ^3,4^**	69	54.58 (14.87)	**56.42 ^1,2,3^**
Time spent outdoors (scale 1–7)	0.032	93	4.77 (1.14)	**4.80 ^1^**	211	5.03 (1.23)	5.02	96	4.81 (1.15)	**4.80 ^2^**	69	5.30 (1.09)	**5.32 ^1,2^**
Participation in organised sports (mins/week)	0.118	88	61.08 (67.09)	**65.78 ^1,2^**	204	44.69 (60.61)	**42.67 ^1^**	89	43.16 (48.52)	**41.22 ^2,3^**	67	49.64 (59.94)	**52.18 ^3^**

Motor competence was measured using the Test of Gross Motor Development (TGMD)-3. Time spent outdoors and participation in organised sports were asked via parental questionnaire. The time spent outdoors was a sum from the time spent outdoors on weekdays and weekends. The scale for the weekdays ranged from 0 to 3 (0 = not at all, 1 = under 30 min/d, 2 = approximately 30–60 min/d and 3 = over 60 min/d), and the scale for the weekends ranged from 0 to 4 (0 = not at all, 1 = under 30 min/d, 2 = approximately 30–60 min/d, 3 = 1–2 h/d and 4 = over 2 h/d). Values are reported as mean and standard deviation (SD) scores. LM skills= locomotor skills, OC skills= object control skills. * Statistically significant difference between residential density areas (adjusted for age, random effect of childcare centre) at the level of *p* < 0.05. Statistically significant differences in bold. In LM skills difference between countryside and metropolitan area: ^1^
*p* = 0.05 *, countryside and cities: ^2^
*p* = 0.025 *, countryside and rural areas: ^3^
*p* = 0.015 * and metropolitan area and rural areas: ^4^
*p* = 0.025 *. In OC skills difference between countryside and metropolitan area: ^1^
*p* = 0.013 *, countryside and cities: ^2^
*p* = 0.015 *, countryside and rural areas: ^3^
*p* = 0.002 **, and rural areas and metropolitan area: ^4^
*p* = 0.015 *. In TGMD-3 total skills difference between countryside and metropolitan area: ^1^
*p* = 0.011 *, countryside and cities: ^2^
*p* = 0.010 **, countryside and rural areas: ^3^
*p* = 0.024 * and rural areas and metropolitan area: ^4^
*p* = 0.010 **. Time spent outdoors, difference between countryside and metropolitan: ^1^
*p* = 0.011 *, and countryside and rural areas: ^2^
*p* = 0.010 **. In participation in organised sports difference between metropolitan and cities: ^1^
*p* = 0.030 *, metropolitan area and rural areas: ^2^
*p* = 0.032 * and countryside and rural areas: ^3^
*p =* 0.030 *.

**Table 6 ijerph-16-01989-t006:** Residential density: differences in boys.

Child Factors	Total Sample	Metropolitan			Cities			Rural Areas			Countryside		
	Overall *p*-Value	*n*	Mean (SD)	Adj. Age	*n*	Mean (SD)	Adj. Age	*n*	Mean (SD)	Adj. Age	*n*	Mean (SD)	Adj. Age
LM skills (max.46p.)	0.094	95	25.68 (8.13)	26.61	210	27.29 (7.82)	26.30	85	24.41 (8.21)	**24.32 ^1^**	83	25.67 (8.56)	**27.21 ^1^**
OC skills (max. 54p.)	0.179	95	24.39 (9.51)	**25.52 ^1^**	210	29.06 (9.12)	27.85	85	26.60 (9.85)	26.49	83	26.86 (9.20)	**28.74 ^1^**
TGMD-3 total (max. 100p.)	0.110	95	50.07 (16.33)	52.13	210	56.35 (15.40)	54.15	85	51.01 (16.62)	**50.51 ^1^**	83	52.53 (15.89)	**55.95 ^1^**
Time spent outdoors (scale 1–7)	0.020	94	4.80 (1.17)	**4.83 ^1,2^**	210	5.40 (1.09)	**5.37 ^1^**	83	5.16 (1.20)	**5.16 ^3^**	83	5.33 (1.07)	**5.38 ^2,3^**
Participation in organised sports (mins/week)	0.939	92	52.49 (66.90)	57.16	200	55.25 (75.62)	50.43	83	46.47 (69.83)	46.14	79	41.27 (59.82)	48.38

Motor competence was measured using the Test of Gross Motor Development (TGMD)-3. Time spent outdoors and participation in organised sports were asked via parental questionnaire. The time spent outdoors was a sum from the time spent outdoors on weekdays and weekends. The scale for the weekdays ranged from 0 to 3 (0 = not at all, 1 = under 30 min/d, 2 = approximately 30–60 min/d and 3 = over 60 min/d), and the scale for the weekends ranged from 0 to 4 (0 = not at all, 1 = under 30 min/d, 2 = approximately 30–60 min/d, 3 = 1–2 h/d and 4 = over 2 h/d). Values are reported as mean and standard deviation (SD) scores. LM skills= locomotor skills, OC skills= object control skills. * Statistically significant difference between residential density areas (adjusted for age, random effect of childcare centre) at the level of *p* < 0.05. Statistically significant differences in bold. In LM skills difference between countryside and rural areas: ^1^
*p* = 0.014 *. In OC skills difference between countryside and metropolitan area: ^1^
*p* = 0.048 *. In TGMD-3 total skills difference between countryside and rural areas: ^1^
*p* = 0.030 *. In time spent outdoors, difference between cities and metropolitan area: ^1^
*p* = 0.006 **, countryside and metropolitan area: ^2^
*p* = 0.012 * and countryside and rural areas: ^3^
*p* = 0.006 **.

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
