# Peer review of "Environmental Correlates of Motor Competence in Children—The Skilled Kids Study"

_ijerph, 2019, doi:10.3390/ijerph16111989_

Round 1

Reviewer 1 Report

This paper was interesting to read, and relevant to consider. I would recommend this research for publication after major revision. First, the authors need to add relevant theory to support their research. Second, the hypotheses need to be re-examined to include the relationship between outdoor play and sports participation on MS. Third, the analyses must include an examination of the relationship between outdoor play and sports participation on MS. Finally, the discussion should be edited to reflect the prior changes. Please see the highlighted sections of the paper with corresponding sticky notes for more details.

Author Response

RESPONSE LETTER

We value the time that you, the editors and reviewers, have taken to review our manuscript, and we thank you for your insightful comments, which have helped us improve our paper. Please find in attached file our point-by-point responses to your comments. The responses to your comments or questions are marked in red. Related modifications to the manuscript have been highlighted in yellow.

Before we proceed to more detailed comments, we would like to point out that we have made some major changes to the manuscript in response to your relevant comments. First, we added the theory of affordances by Gibson (1977) to the introduction. Second, we reduced the geographical location categories from four to three, as there were no geographical differences in terms of temperature or the amount of daylight between the categories ‘metropolitan area’ and ‘Southern Finland’. There is now only one Southern Finland category (see table 1) in geographical location and four remaining in terms of residential density (metropolitan area, cities, rural areas and countryside). Third, the correlations between LM, OC skills, TGMD-3 and time spent outdoors and participation in organised sports are analysed and included in the manuscript. Finally, we switched to using the linear mixed-effects model rather than partial correlation in order to analyse the differences between groups of geographical locations (Southern, Central and Northern Finland) and residential density (metropolitan area, cities, rural areas and countryside) in MC (LM, OC skills and TGMD-3), the time spent outdoors and participation in organised sports (Tables 3a–4b). The Results were accordingly rewritten and the Discussion section is slightly different. To conclude, several new references are included in the revised manuscript.

Response to Reviewer 1 Comments in attached file. 

Reviewer 2 Report

Introduction

Page 1, lines 37-39: Change the first part of this sentence, “Campos et al. [9] suggest that improved locomotor (LM) skills provide more numerous and variable social and cognitive experiences and support their development during infancy;” to read as, “Campos et al. [9] suggest that improved locomotor (LM) skills provide more numerous and variable social and cognitive experiences and support the development of such experiences during infancy;”

Page 2, lines 90-93 (last two sentences of the Introduction): Need clarity here.  The first sentence proposes three variables (physical environment, time outdoors, and participation in organized sports) as possibly related to MS.  The next sentence (the hypothesis) states that only one of these variables (physical environment) would be related to MS.  Therefore, could the hypothesis be stated something like, “We hypothesized that the physical environment would have the strongest association with children’s MS.”  Additionally, to help tie the literature review to the hypothesis, I suggest adding a clause at the beginning of this sentence.  The clause might read something like, “Given the bidirectional connection between MS and PA and between PA the environment, we hypothesized that children’s the physical environment would have the strongest association with children’s MS.”

Methods

Page 4, line 136: How many educated observers (i.e. skill raters) were there in the study? One at each location?

Page 4, line 144: The interrater reliability was based on how many different raters?

Page 4, lines 154 and 159: Is test-retest validity or test-retest reliability?  How many measurements were used to calculate the test-retest?

Results

Page 5, line 181: Who are the “respondents”?  Are they the parents who responded to the time spent outdoors and participation in organized sports questionnaire? Please clarify.

Table 2: BMI SDS scores do not make sense – why not just report the mean BMI and standard deviation (SD)?

Table 2: Please report the p-values for the gender differences for underweight, normal weight, overweight, and obese.  I suggest reporting all information for Table 2 (i.e., all minimums and maximums, all overall means and SDs, and all p-values).  

Author Response

RESPONSE LETTER

We value the time that you, the editors and reviewers, have taken to review our manuscript, and we thank you for your insightful comments, which have helped us improve our paper. Please find in attached file our point-by-point responses to your comments. The responses to your comments or questions are marked in red. Related modifications to the manuscript have been highlighted in yellow.

Before we proceed to more detailed comments, we would like to point out that we have made some major changes to the manuscript in response to your relevant comments. First, we added the theory of affordances by Gibson (1977) to the introduction. Second, we reduced the geographical location categories from four to three, as there were no geographical differences in terms of temperature or the amount of daylight between the categories ‘metropolitan area’ and ‘Southern Finland’. There is now only one Southern Finland category (see table 1) in geographical location and four remaining in terms of residential density (metropolitan area, cities, rural areas and countryside). Third, the correlations between LM, OC skills, TGMD-3 and time spent outdoors and participation in organised sports are analysed and included in the manuscript. Finally, we switched to using the linear mixed-effects model rather than partial correlation in order to analyse the differences between groups of geographical locations (Southern, Central and Northern Finland) and residential density (metropolitan area, cities, rural areas and countryside) in MC (LM, OC skills and TGMD-3), the time spent outdoors and participation in organised sports (Tables 3a–4b). The Results were accordingly rewritten and the Discussion section is slightly different. To conclude, several new references are included in the revised manuscript.

Response to Reviewer 2 Comments in attached file.

Reviewer 3 Report

The current manuscript aims to analyse the environment correlates with motor competence in childhood. From a methods perspective, the majority of the aspects can be seen as strengths such as an appropriate analysis of the validity and reliability of the TGMD-3 test to assess children's motor skill competence. As a result, this is an interesting study.

The major issues are related to the theoretical background, what the authors mean about “environment” and the statistical analyses conducted.

Through the introduction section the authors seem to be jumping the content without clear connection (e.g., cognitive function, role of the curriculum or preschoolers sitting time have been included as a part of the introduction but the connection with motor skills and environment is not clear enough). Please, note that a thorough re-elaboration of the rationale of the introduction and removal of the "gaps" is required. In this line, I would suggest the authors provide theoretical background in which the content can be based on. That is, on the one hand, considering the points related to environment and curriculum a model-based approach such as non-linear pedagogy could be interesting. On the other hand, taking into account cognitive function, environment and motor skill learning, a constraint led-approach could also provide an insight into the introduction section.

Do the authors refer to the level of motor skills or just to the fact of practising motor skills? Depending on the sense the authors would like to express, the terminology should be modified. For instance, in the case the authors refer to the level of motor skill it would be recommended they refer to "motor competence".

Regarding the statistical analyses, taking into account the purpose of the current study, why did not the authors conduct mixed-model regression analyses (e.g., multilevel mixed-effects linear regression models)? With this suggested approach, the authors could adjust the analyses and provide the environmental correlates that influence children's motor skills competence appropriately. 

Specific comments

Title. It is recommended the authors change the title to "Leisure, physical and geographical environmental correlates of motor competence in children - the Skilled Kid Study"

Line 45. The authors refer to the association between locomotor skills and development during infancy. However, there is not mention to the object control skills and the influence of the environment in children's development. This is important because object control skills are measured by the TGMD-3 as explained in the methods.

Line 49. The content in the paragraph seems to be beyond the aim of this manuscript. At least lines 49-52 should be removed. The rest of the content could be integrated in a rewriten introduction.

Lines 68-69. Please, provide scientific support to this statement.

Lines 122-124. This decision requires further justification. It would be nice to provide scientific evidences for this matter.

Line 166. Statistical analyses subsection. See my point in the general comments.

Line 243. Once mixed-models are conducted, the discussion should be rewritten accordingly.

Lines 266-268. This is one reason why I recommended providing theoretical background into the introduction section. This is the first time "the theory of affordances" is stated but not explanation is provided previously. It is suggested that in the introduction section, the authors summarize what this theory refers to.

Author Response

RESPONSE LETTER

We value the time that you, the editors and reviewers, have taken to review our manuscript, and we thank you for your insightful comments, which have helped us improve our paper. Please find in attached file our point-by-point responses to your comments. The responses to your comments or questions are marked in red. Related modifications to the manuscript have been highlighted in yellow.

Before we proceed to more detailed comments, we would like to point out that we have made some major changes to the manuscript in response to your relevant comments. First, we added the theory of affordances by Gibson (1977) to the introduction. Second, we reduced the geographical location categories from four to three, as there were no geographical differences in terms of temperature or the amount of daylight between the categories ‘metropolitan area’ and ‘Southern Finland’. There is now only one Southern Finland category (see table 1) in geographical location and four remaining in terms of residential density (metropolitan area, cities, rural areas and countryside). Third, the correlations between LM, OC skills, TGMD-3 and time spent outdoors and participation in organised sports are analysed and included in the manuscript. Finally, we switched to using the linear mixed-effects model rather than partial correlation in order to analyse the differences between groups of geographical locations (Southern, Central and Northern Finland) and residential density (metropolitan area, cities, rural areas and countryside) in MC (LM, OC skills and TGMD-3), the time spent outdoors and participation in organised sports (Tables 3a–4b). The Results were accordingly rewritten and the Discussion section is slightly different. To conclude, several new references are included in the revised manuscript.

Response to Reviewer 3 Comments in attached file.

Round 2

Reviewer 1 Report

The edits have helped to refine and clarify the research questions and associated results. The addition of the theoretical framework strengthened the research. Finally, narrowing the geographic areas to three makes the analysis much easier to understand and interpret. This revision is much improved and I now recommend the paper for publication.